# Maternal dietary folate intake with folic acid supplements and wheeze and eczema in children aged 2 years in the Japan Environment and Children's Study

Hideyuki Masuda[1], Sumitaka Kobayashi[1], Chihiro Miyashita[1], Sachiko Itoh[1], Yu Ait Bamai[1], Yasuaki Saijo[2], Yoshiya Ito[3], Reiko Kishi[1], Atsuko Ikeda-Araki[1,4]*, Japan Environment and Children's Study (JECS) Group[¶]

1 Center for Environmental and Health Sciences, Hokkaido University, Sapporo, Japan, 2 Department of Social Medicine, Asahikawa Medical University, Asahikawa, Japan, 3 Faculty of Nursing, Japanese Red Cross Hokkaido College of Nursing, Kitami, Japan, 4 Faculty of Health Sciences, Hokkaido University, Sapporo, Japan

¶ Membership of the Bunny Genome Sequencing Consortium is provided in the Acknowledgments
* AAraki@cehs.hokudai.ac.jp

**Data Availability Statement:** The data are unsuitable for public deposition owing to ethical restrictions and the legal framework of Japan. It is

## Abstract

Maternal intake of folic acid supplements is reportedly associated with the risk of early-onset allergies in offspring. However, only a few studies have considered the intake of both folic acid supplements and dietary folate. Here, the relationship between maternal intake of folic acid supplements and allergic symptoms such as wheeze and eczema in offspring was analyzed while considering dietary folate intake. We examined 84,361 mothers and 85,114 children in the Japan Environment and Children's Study. The participants were divided into three groups depending on maternal folic acid supplementation ("no use," "occasional use," and "daily use"). Each group was then subdivided into three groups based on total folic acid and dietary folate intake. Outcomes were determined considering the wheeze and eczema status of each child at the age of 2 years. The status was based on the International Study of Asthma and Allergies in Childhood. It was found that 22.1% of the mothers took folic acid supplements daily. In contrast, 56.3% of the mothers did not take these supplements. Maternal intake of folic acid supplements was not associated with wheeze and eczema in the offspring. In contrast, only dietary folate intake was positively associated with wheeze at the age of 2 (adjusted odds ratio, 1.103; 95% confidence interval, 1.003–1.212). However, there is no scientific evidence of a biological mechanism that clarifies this result. Potential confounders such as other nutrition, outdoor/indoor air pollution, and genetic factors may have affected the results. Therefore, further studies on the association between maternal intake of folic acid and allergic symptoms at the age of 3 or above are needed to confirm the results of this study.

Trial registration

UMIN Clinical Trials Registry (number: UMIN000030786)

prohibited by the Act on the Protection of Personal Information (Act No. 57 of 30 May 2003, amendment on 9 September 2015) to publicly deposit data containing personal information. Ethical Guidelines for Medical and Health Research Involving Human Subjects enforced by the Japan Ministry of Education, Culture, Sports, Science and Technology and the Ministry of Health, Labour and Welfare also restricts the public sharing of epidemiologic data. All inquiries about access to data should be addressed to Dr Shoji F. Nakayama, JECS Programme Office, National Institute for Environmental Studies (jecs-en@nies.go.jp).

**Funding:** This study was funded by the Ministry of Environment, Japan. The funders had no role in study design, data collection and analysis, decision to publish, or preparation of the manuscript.

**Competing interests:** The authors have declared that no competing interests exist.

## Introduction

Folic acid, a B vitamin, plays an important role in erythropoiesis and DNA methylation. It also decreases the risk of neural tube defects [1–3]. Folic acid deficiency (serum folic acid < 5 ng/mL) causes serious health impairments, including neuropsychiatric disorders [4]. Hence, 68 countries have a mandatory folic acid fortification in food [5]. However, Japan has no such mandate. The dietary folate intake recommendation by the Ministry of Health, Labour and Welfare in Japan for adults is 240 μg/day. Moreover, to reduce the risk of neural tube closure in the fetus, women who are in the pre-pregnancy stage and in the first trimester of pregnancy are recommended to take in 400 μg/day folic acid derived from supplements [6]. In addition, dietary folate intake (240 μg/day) is recommended in the second/third trimester of pregnancy as additional intake [6]. However, in a Japanese study from 2017, only 45.1% of pregnant women used folic acid supplements [7]. Moreover, on average, dietary folate intake is 243 μg/day in Japanese pregnant women [8]. Thus, the current recommendations for the intake of folic acid supplements and dietary folate during pregnancy in Japan are insufficient. Dietary folate and supplementary folic acid differ in their chemical structures and contain polyglutamic and monoglutamic acid, respectively. The major difference between them is their absorption rate in the digestive tract during the metabolism of polyglutamic to monoglutamic acid [9].

Allergies such as asthma and atopic dermatitis are major public health problems worldwide. A nationwide cohort study in Japan showed that approximately 13%–14% of children at the age of 3 suffer from wheeze or eczema [10]. Maternal risk factors for allergies have been reported by some studies; these include maternal gestational smoking, maternal stressful events, and maternal obesity [11–13]. Additionally, cohort studies have reported that folic acid supplementation during pregnancy is a risk factor for allergies in offspring [14–18]. In the Norwegian Mother and Child Cohort Study, folic acid supplementation in the first trimester of pregnancy increased the relative risk (RR) of wheeze (RR, 1.06; 95% confidence interval [CI], 1.03–1.10) compared with no supplementation [14]. In the Prevention and Incidence of Asthma and Mite Allergy birth cohort study, Bekkers et al. found that the risk of contracting wheeze in children at 1 year of age (prevalence ratio, 1.20; 95% CI, 1.04–1.39) was higher with the use of folic acid-containing supplements than with the use of folic acid-lacking supplements [15]. Moreover, in the Generation R study, a folate concentration of ≥16.21 nmol/L in plasma during pregnancy increased the risk of atopic dermatitis in offspring (odds ratio [OR], 1.16; 95% CI, 1.03–1.32) compared with lower concentrations (≤10.30 nmol/L) [16].

Nevertheless, only a few studies have analyzed the association between folic acid supplementation and allergies in offspring considering dietary folate intake. In the Generation 1 Cohort Study in Australia, high maternal intake of folic acid supplements during late pregnancy was associated with a higher risk of asthma in children at 3.5 years of age (RR, 1.26; 95% CI, 1.09–1.47) compared with no supplement intake. However, maternal intake of dietary folate was not associated with asthma in offspring [17]. Furthermore, Parr et al. considered total folic acid intake (dietary and supplemental) and observed that a folate-rich diet combined with at least 400 μg/day folic acid supplement intake (total ≥ 578 μg/day) increased the relative risk of childhood asthma (RR,1.23; 95% CI, 1.06–1.44) compared with low intake (total ≤ 146 μg/day) in the Norwegian Mother and Child Cohort Study [18]. Dietary folate and supplemental folic acid are absorbed as monoglutamate and converted to polyglutamate in tissue [9]; there is no significant difference in their *in-vivo* functions. Therefore, the effects of dietary folate intake must be considered to evaluate the contribution of folic acid supplementation during pregnancy to allergic risk in offspring.

In previous studies, allergic symptoms such as asthma, wheeze, and atopic dermatitis in offspring were associated with the maternal intake of folic acid [14–18]. In this study, we

evaluated the association between folic acid supplements and dietary folate intake during pregnancy and wheeze and eczema, which are early symptoms of asthma and atopic dermatitis in offspring.

## Materials and methods

The Japan Environment and Children's Study (JECS) protocol was reviewed and approved by the Institutional Review Board on Epidemiological Studies of the Ministry of the Environment and the Ethics Committees of all participating institutions. Details of the JECS have been described previously [19,20]. In brief, the JECS is a nationwide birth cohort study in Japan that elucidated environmental factors that affect the health and development of children. The JECS recruited approximately 100,000 participants during pregnancy between January 2011 and March 2014. Written informed consent was obtained from parents or guardians. In this study, we used the jecs-ta-20190930 dataset, which contains data on 104,062 fetal records. In total, 18,948 records were excluded for the following reasons: stillbirth; miscarriage; unanswered birth questions; and non-responses to questionnaires during the second/third trimester, questionnaires regarding children aged 2 years, questionnaires about folic acid intake (supplemental or dietary), and queries regarding both symptoms (wheeze and eczema). Finally, data from 85,114 records were used in the analysis (Fig 1).

Exposure was defined as the self-reported use of folic acid supplements and dietary folate intake in the past month obtained from a questionnaire distributed during the second/third trimester. The intake of folic acid supplements was divided into three groups: mothers who did not take folic acid supplements were included in the "no use" group; mothers who occasionally took the supplements were included in the "occasional use" group; and mothers who took them at least once a day were included in the "daily use" group. Dietary folate intake (μg/day) after conception was calculated using a semiquantitative food frequency questionnaire

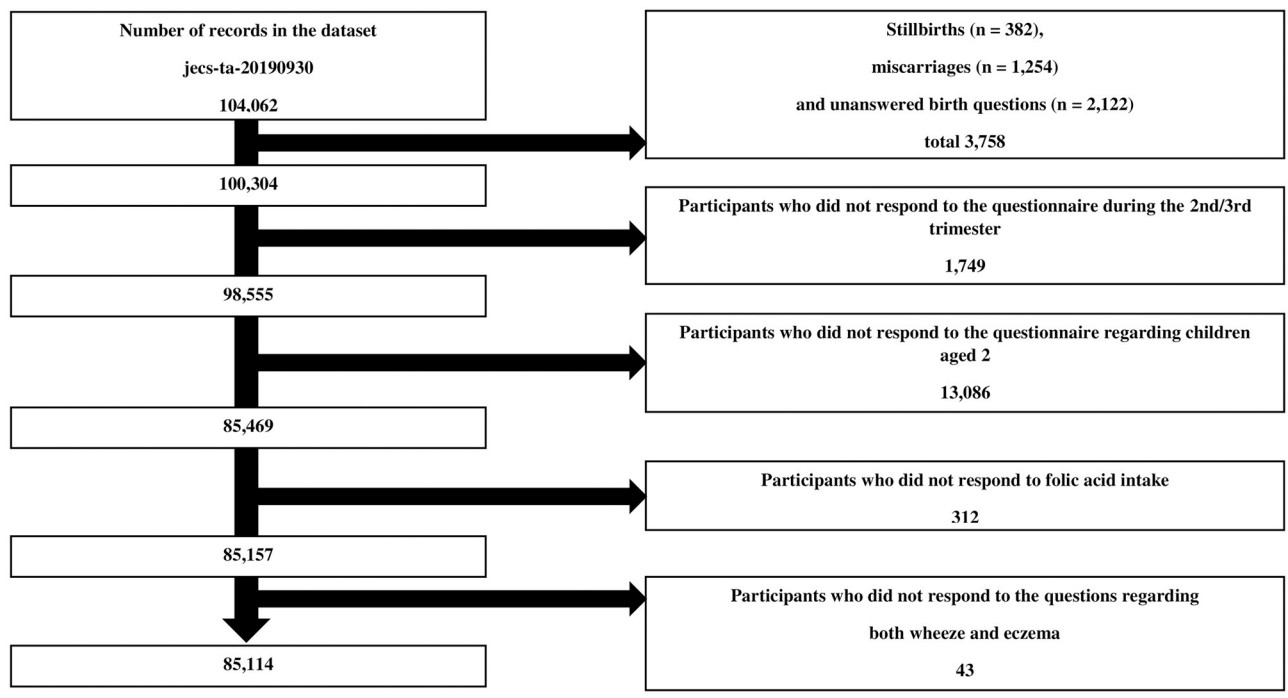

**Fig 1. Flow chart of the participant selection.**

(FFQ) [21]. In brief, the FFQ asked the participants about the frequency of consumption of 172 food and beverage items, from never to seven or more times per day for food, and ten or more glasses per day for beverages. Thereafter, the intake of 53 nutrients was calculated. The folic acid intake (μg) was divided into three groups (<240, 240–479, and ≥480 μg/day) based on the standards issued by the Ministry of Health, Labour and Welfare of Japan and a previous report from the JECS [6,22].

Outcomes were wheeze and eczema in offspring at the age of 2. These were defined using a questionnaire based on the ISAAC [23–25]. A partially modified version of the validated Japanese ISAAC questionnaire for children aged 6–7 years was used [10]. Wheeze was defined as a positive response to the following questions: "Have you had wheezing or whistling in the chest at any time in the past?" and "Have you had wheezing or whistling in the chest in the last 12 months?". Eczema was defined as a positive response to the following questions: "Have you ever had a recurring itchy rash for at least 6 months? If yes: Have you had this itchy rash at any time in the last 12 months? If yes: Has this itchy rash at any time affected any of the following places: the folds of the elbows, behind the knees, in front of the ankles, under the buttocks, or around the neck, ears, or eyes?".

Maternal and offspring characteristics, such as folic acid intake (supplemental and dietary) and wheeze and eczema symptoms were examined. First, we examined the associations with folic acid supplement and dietary folate intake separately. For folic acid supplement intake, "no use" was selected as a reference, whereas for dietary folate intake, "240–479 μg/day" was selected as the reference range based on the recommended consumption of folate per day for individuals during the first trimester (240 μg/day) and second/third trimester (480 μg/day) of pregnancy [6]. Second, to examine the interaction of maternal folic acid supplementation and dietary folate intake, we combined folic supplementation and dietary folate intake to find the association with symptoms among offspring. In this analysis, "no use" and" 240–479 μg/day" were selected as the reference [8]. The ORs and their 95% CIs were determined using crude and multivariate logistic regression analyses. To analyze the adjusted OR (aOR), multivariate logistic regression analyses were performed after adjusting for maternal age (≤24, 25–34, and ≥35 years), sex (male/female), gestational age (<37, ≥37 to <42, and ≥42 weeks), maternal and paternal education (junior high school/high school/technical junior college, technical [vocational] college/associate degree/bachelor's degree/graduate degree [Master's/Doctorate]), history of maternal allergy (yes/no), maternal smoking during pregnancy (no smoking/passive smoking/active smoking), maternal alcohol consumption during pregnancy (yes/no), maternal body mass index (BMI) before pregnancy (<18.5, ≥18.5 to <25.0, ≥25.0 to <30.0, and ≥30.0), parity (primipara/multipara), breastfeeding term (0 months, ≥1.00 to <6.00 months, ≥6.00 to <13.00 months, and ≥13.00 months), and nursery school and day-care center for children aged 1 or 2 years (yes/no). These factors were selected based on previous studies that reported an association between folic acid intake during pregnancy and allergies in offspring [14–18]. All analyses were performed using SPSS Statistics version 22 (IBM Corp., Armonk, NY, USA).

## Results

Table 1 shows the characteristics of the 84,361 mothers. Mothers in the "daily use" group constituted 22.1% of all participants, whereas those in the "no use" group equaled 56.3%. Mothers in the "daily use" group exhibited the following parameters more often than those in the "no use" group: older maternal age, high parental education, no smoking, no alcohol consumption, existing allergies, singleton, and appropriate BMI. Compared with those in the "<240 μg/day" group, mothers in the "240–479 μg/day" group (40.6% of all participants) more often exhibited

**Table 1. Characteristics of the participants (mother).**

| | | All participants (total = 84,361 mothers) | | Folic acid supplement n (%) | | | Folate (µg) diet per day n (%) | | |
|---|---|---|---|---|---|---|---|---|---|
| | | n | % | No use | Occasional use | Daily use | <240 | 240–479 | ≥480 |
| **Maternal age** | **Mean (SD)** | 31.4 (4.9) | | | | | | | |
| | **≤24 years** | 7,168 | 8.5 | 4,911 (10.3%) | 1,381 (7.6%) | 876 (4.7%) | 4,884 (10.9%) | 1,971 (5.8%) | 313 (6.0%) |
| | **25–34 years** | 53,570 | 63.5 | 30,194 (63.6%) | 11,872 (65.1%) | 11,504 (61.8%) | 29,115 (64.9%) | 21,357 (62.4%) | 3,098 (59.0%) |
| | **≥35 years** | 23,622 | 28 | 12,397 (26.1%) | 4,976 (27.3%) | 6,249 (33.5%) | 10,894 (24.3%) | 10,891 (31.8%) | 1,837 (35.0%) |
| | **No answer** | 1 | <0.01 | | | | | | |
| **Education (mother)** | **Junior high/high school/technical college** | 30,182 | 35.8 | 18,477 (39.0%) | 5,820 (32.1%) | 5,885 (31.7%) | 17,911 (40.1%) | 10,591 (31.0%) | 1,680 (32.1%) |
| | **Professional school/junior college/ university/graduate school** | 53,864 | 63.8 | 28,848 (61.0%) | 12,333 (67.9%) | 12,683 (68.3%) | 26,791 (59.9%) | 23,523 (69.0%) | 3,550 (67.9%) |
| | **No answer** | 315 | 0.4 | | | | | | |
| **Education (father)** | **Junior high/high school/technical college** | 37,315 | 44.2 | 22,678 (48.2%) | 7,379 (40.8%) | 7,258 (39.3%) | 21,066 (47.4%) | 13,988 (41.2%) | 2,261 (43.5%) |
| | **Professional school/junior college/ university/graduate school** | 46,291 | 54.9 | 24,358 (51.8%) | 10,702 (59.2%) | 11,231 (60.7%) | 23,357 (52.6%) | 19,992 (58.8%) | 2,942 (56.5%) |
| | **No answer** | 755 | 0.9 | | | | | | |
| **Allergy (mother)** | **No** | 35,964 | 42.6 | 21,021 (44.4%) | 7,361 (40.5%) | 7,582 (40.9%) | 19,589 (43.8%) | 14,184 (41.6%) | 2,191 (41.9%) |
| | **Yes** | 48,081 | 57 | 26,307 (55.6%) | 10,796 (59.5%) | 10,978 (59.1%) | 25,154 (56.2%) | 19,893 (58.4%) | 3,034 (58.1%) |
| | **No answer** | 316 | 0.4 | | | | | | |
| **Smoking (during pregnancy)** | **No smoking** | 45,386 | 53.8 | 24,138 (51.7%) | 10,182 (56.9%) | 11,066 (60.3%) | 22,821 (51.8%) | 19,693 (58.2%) | 2,972 (57.6%) |
| | **Passive smoking** | 34,410 | 40.8 | 20,438 (43.8%) | 7,144 (39.9%) | 6,828 (37.2%) | 19,271 (43.7%) | 13,130 (39.0%) | 2,009 (38.9%) |
| | **Active smoking** | 3,096 | 3.7 | 2,069 (4.4%) | 570 (3.2%) | 457 (2.5%) | 1,960 (4.4%) | 958 (2.8%) | 178 (3.5%) |
| | **No answer** | 1,469 | 1.7 | | | | | | |
| **Alcohol consumption (during pregnancy)** | **No** | 81,447 | 96.5 | 45,647 (96.8%) | 17,667 (97.6%) | 18,133 (98.0%) | 43,420 (97.5%) | 32,991 (97.0%) | 5,036 (96.5%) |
| | **Yes** | 2,302 | 2.7 | 1,497 (3.2%) | 430 (2.4%) | 375 (2.0%) | 1,113 (2.5%) | 1,008 (3.0%) | 181 (3.5%) |
| | **No answer** | 612 | 0.7 | | | | | | |
| **Parity** | **Primipara** | 35,485 | 42.1 | 17,043 (36.1%) | 8,612 (47.8%) | 9,830 (53.4%) | 20,633 (46.4%) | 13,142 (38.7%) | 1,710 (32.8%) |
| | **Multipara** | 48,129 | 57.1 | 30,129 (63.9%) | 9,422 (52.2%) | 8,578 (46.6%) | 23,829 (53.6%) | 20,804 (61.3%) | 3,496 (67.2%) |
| | **No answer** | 747 | 0.9 | | | | | | |

(*Continued*)

**Table 1.** (Continued)

| | | All participants | | Folic acid supplement n (%) | | | Folate (µg) diet per day n (%) | | |
|---|---|---|---|---|---|---|---|---|---|
| | | (total = 84,361 mothers) | | | | | | | |
| | | n | % | No use | Occasional use | Daily use | <240 | 240–479 | ≥480 |
| Body mass index (before pregnancy) | <18.5 | 13,644 | 16.2 | 7,582 (16.0%) | 2,967 (16.3%) | 3,095 (16.6%) | 7,420 (16.5%) | 5,442 (15.9%) | 782 (14.9%) |
| | ≥18.5 and <25.0 | 62,187 | 73.7 | 3,4678 (73.0%) | 13,608 (74.7%) | 13,901 (74.6%) | 32,766 (73.0%) | 25,537 (74.6%) | 3,884 (74.0%) |
| | ≥25.0 and <30.0 | 6,568 | 7.8 | 3,961 (8.3%) | 1,304 (7.2%) | 1,303 (7.0%) | 3,601 (8.0%) | 2,514 (7.3%) | 453 (8.6%) |
| | ≥30.0 | 1,950 | 2.3 | 1,276 (2.7%) | 347 (1.9%) | 327 (1.8%) | 1,100 (2.5%) | 723 (2.1%) | 127 (2.4%) |
| | No answer | 12 | 0.01 | | | | | | |
| Total | | 84,361 | 100 | 47,503 (56.3%) | 18,229 (21.6%) | 18,629 (22.1%) | 44,894 (53.2%) | 34,219 (40.6%) | 5,248 (5.2%) |

SD, standard deviation.

older maternal age, high maternal and paternal education, no smoking, existing allergies, primipara, and an appropriate BMI. However, a high frequency of alcohol consumption was exhibited by the "240–479 µg/day" group.

Table 2 presents the characteristics of the offspring. Among all children, 24.0% had wheeze and 13.0% had eczema. Male sex, premature birth, high birth weight, short-term breastfeeding, children commuting to nursery schools and day-care centers, and children without pets exhibited a high wheeze frequency. Furthermore, eczema was more prevalent in the male sex, with high birth weight, long-term breastfeeding, a regular commute to a day-care center, and ownership of pets. The characteristics of the mother were closely linked with a high frequency of symptoms in the offspring. Offspring with low paternal education, maternal allergies, as well as those whose mothers exhibited high alcohol consumption, multipara, and high BMI generally exhibited a high wheeze frequency. In addition, offspring with high maternal and paternal education, maternal allergies, as well as those whose mothers exhibited high alcohol consumption and multipara exhibited a high eczema frequency (S1 Table).

The association between folic acid intake and wheeze or eczema in children was assessed using logistic regression analysis (Table 3). The risk of wheeze increased in the "≥480 µg/day" group compared with that in the "240–479 µg/day" group (aOR 1.113; 1.037–1.194). In addition, the risk of wheeze decreased in the "<240 µg/day" group compared with that in the "240–479 µg/day" group (aOR 0.942; 0.909–0.977). The risk of eczema in offspring was low when the intake of dietary folate was low (<240 µg/day) compared with the risk when the intake of dietary folate was 240–479 µg/day (aOR 0.915; 0.876–0.956). However, the risk of eczema did not increase in the "≥480 µg/day" group compared with that in the "240–479 µg/day" group (aOR 1.009; 0.925–1.101).

Finally, the association between the combined folic acid supplemental/dietary folic acid intake and wheeze or eczema in children was analyzed (Table 4). In the "no use" group, the OR of wheeze varied depending on the dietary folate intake ("<240 µg/day": aOR 0.943; 0.900–0.988; "≥480 µg/day": aOR 1.103; 1.003–1.212). Moreover, daily use of folic acid supplement at a concentration of <240 µg/day decreased the risk of wheeze (aOR 0.906; 0.849–0.966). Regarding eczema, no intake of folic acid supplements or intake of these supplements

**Table 2. Characteristics of the participants (children).**

| | | All participants | | Wheeze | | Eczema | |
|---|---|---|---|---|---|---|---|
| | | (total = 85114 children) | | n (%) | | n (%) | |
| | | n | % | Yes | No | Yes | No |
| **Sex** | **Male** | 43609 | 51.2 | 11633 (57.0%) | 31698 (49.4%) | 6207 (56.0%) | 37222 (50.5%) |
| | **Female** | 41504 | 48.8 | 8785 (43.0%) | 32467 (50.6%) | 4878 (44.0%) | 36471 (49.5%) |
| | **No answer** | 1 | 0 | | | | |
| **Gestational age (months)** | <37 | 4452 | 5.2 | 1299 (6.4%) | 3177 (4.9%) | 550 (5.0%) | 3884 (5.3%) |
| | ≥37 and <42 | 80313 | 94.4 | 19040 (93.4%) | 60784 (94.9%) | 10497 (94.9%) | 69500 (94.5%) |
| | ≥42 | 189 | 0.2 | 39 (0.2%) | 147 (0.2%) | 18 (0.2%) | 171 (0.2%) |
| | No answer | 160 | 0.2 | | | | |
| **Multiple birth** | singleton | 83598 | 98.2 | 20039 (98.1%) | 63037 (98.2%) | 10880 (98.1%) | 72388 (98.2%) |
| | multitone | 1516 | 1.8 | 380 (1.9%) | 1128 (1.8%) | 206 (1.9%) | 1305 (1.8%) |
| **Birth weight (g)** | <2500 | 7559 | 8.9 | 1916 (9.4%) | 5593 (8.7%) | 895 (8.1%) | 6639 (9.0%) |
| | ≥2500 and <4000 | 76624 | 90 | 18279 (89.8%) | 57873 (90.4%) | 10043 (90.8%) | 66274 (90.1%) |
| | ≤4000 | 724 | 0.9 | 171 (0.8%) | 548 (0.9%) | 117 (1.1%) | 605 (0.8% |
| | No answer | 207 | 0.2 | | | | |
| **Birth year** | **2011** | 8313 | 9.8 | 2068 (10.1%) | 6195 (9.7%) | 1033 (9.3%) | 7230 (9.8%) |
| | **2012** | 24055 | 28.3 | 5693 (27.9%) | 18212 (28.4%) | 2971 (26.8%) | 20997 (28.5%) |
| | **2013** | 30130 | 35.4 | 7312 (35.8%) | 22632 (35.3%) | 4030 (36.4%) | 25989 (35.3%) |
| | **2014** | 22616 | 26.6 | 5436 (26.2%) | 17126 (26.7%) | 3052 (27.5%) | 19477 (26.4%) |
| **Breastfeeding term (months)** | **0** | 2099 | 2.5 | 513 (2.6%) | 1573 (2.5%) | 232 (2.1%) | 1857 (2.6%) |
| | ≥1.00 and <6.00 | 14588 | 17.1 | 3866 (19.3%) | 10605 (16.8%) | 1685 (15.5%) | 12837 (17.7%) |
| | ≥6.00 and <13.00 | 52824 | 62.1 | 12543 (62.6%) | 39965 (63.3%) | 6904 (63.4%) | 45737 (63.1%) |
| | ≥13.00 | 14158 | 16.6 | 3107 (15.5%) | 10982 (17.4%) | 2074 (19.0%) | 12015 (16.6%) |
| | No answer | 1445 | 1.7 | | | | |
| **Nursery school and day care center for children** | **No** | 40974 | 48.1 | 6028 (29.9%) | 34774 (55.0%) | 4935 (45.1%) | 35870 (49.4%) |
| | **Yes** | 42937 | 50.4 | 14156 (70.1%) | 28432 (45.0%) | 5996 (54.9%) | 36787 (50.6%) |
| | **No answer** | 1203 | 1.4 | | | | |
| **History of pet ownership (~1.5 y)** | **No** | 69199 | 81.3 | 16397 (82.9%) | 52379 (83.6%) | 9069 (84.1%) | 59871 (83.4%) |
| | **Yes** | 13725 | 16.1 | 3373 (17.1%) | 10267 (16.4%) | 1709 (15.9%) | 11952 (16.6%) |
| | **No answer** | 2190 | 2.6 | | | | |
| **Total** | | 85,114 | 100 | 20,419 (24.0) | 64,165 (75.4) | 11,086 (13.0) | 73,693 (86.6) |

at a concentration of <240 μg/day decreased the risk of eczema compared with "240–479 μg/day" (aOR 0.880; 0.830–0.933). Moreover, "daily use and <240 μg/day" decreased the risk of eczema (aOR 0.914; 0.845–0.988). Collectively, the lack of dietary folate intake decreased the ORs of symptoms. However, maternal folic acid supplementation did not alter the risk of symptoms.

## Discussion

We observed that high maternal intake of dietary folate was a risk factor for wheeze and eczema in children at 2 years of age. Conversely, folic acid supplements were not associated with wheeze and eczema in the offspring. To the best of our knowledge, this is the first study in Japan to demonstrate the association between maternal intake of folic acid supplements and allergic symptoms in offspring while considering dietary folate intake separately.

Previous studies have reported various results. Bekkers et al. defined exposure as folic acid supplement intake during pregnancy (yes/no). Supplement intake increased the risk of wheeze

**Table 3. Exposure to folic acid supplement or dietary folate and allergy.**

| | | n | % | Wheeze | | | | Eczema | | | |
|---|---|---|---|---|---|---|---|---|---|---|---|
| | | | | Crude[a] | | Adjusted[b] | | Crude[a] | | Adjusted[b] | |
| | | | | OR | 95% CI | OR | 95% CI | OR | 95% CI | OR | 95% CI |
| Folic acid supplement | No use | 47,894 | 56.3 | Ref | | Ref | | Ref | | Ref | |
| | Occasional use | 18,368 | 21.6 | 0.975 | 0.937–1.014 | 1.019 | 0.976–1.063 | 1.091 | 1.038–1.147* | 1.078 | 1.023–1.136* |
| | Daily use | 18,852 | 22.1 | 0.874 | 0.839–0.909* | 0.975 | 0.933–1.018 | 1.008 | 0.958–1.060 | 1.016 | 0.963–1.071 |
| Dietary folate (µg/day) | <240 | 45,260 | 53.2 | 0.946 | 0.915–0.978* | 0.942 | 0.909–0.977* | 0.89 | 0.854–0.928* | 0.915 | 0.876–0.956* |
| | 240–479 | 34,540 | 40.6 | Ref | | Ref | | Ref | | Ref | |
| | ≥480 | 5,314 | 6.2 | 1.149 | 1.076–1.227* | 1.113 | 1.037–1.194* | 1.019 | 0.938–1.108 | 1.009 | 0.925–1.101 |

CI, confidence interval; OR, odds ratio.

[a]Crude is non-adjusted.

[b]Adjusted is adjusted for maternal age, sex, gestational age, education (mother and father), allergy (mother), smoking (during pregnancy), alcohol consumption (during pregnancy), body mass index (before pregnancy), parity, breastfeeding term, nursery school and day-care center for children.

in the first year [15]. Haberg et al. defined exposure as maternal supplementation before/after the 12th week (yes/no) and observed that supplementation increases the risk of wheeze in the first trimester of pregnancy [14]. Kiefte-de Jong et al. defined exposure as supplement intake in the first trimester (yes/no) and observed no association [16]. All these studies evaluated the association between maternal intake of folic acid supplement only and allergies. However, the periods when data were collected differed among these studies. Two out of three studies showed the association between maternal folic acid intake and allergies in offspring [14–16]. Moreover, Parr et al. considered the total amount of folic acid and showed that high folic acid intake increases the relative risk of asthma in children compared with low intake [18]. They did not show the influence by each folic acid supplement and dietary folate. Whitrow et al. considered folic acid and folate intake as continuous variables similar to exposure and showed an association between high maternal folic acid supplement intake and allergies. The exposure in the study by Whitrow et al. is the most similar to that in our study [17]. However,

**Table 4. Folic acid (supplement and diet) and allergies.**

| | | | | Wheeze | | | | Eczema | | | |
|---|---|---|---|---|---|---|---|---|---|---|---|
| | | | | Crude[a] | | Adjusted[b] | | Crude[a] | | Adjusted[b] | |
| Folic acid supplement | Dietary folate (µg/day) | n | (%) | OR | 95% CI | OR | 95% CI | OR | 95% CI | OR | 95% CI |
| No use | <240 | 26,227 | 30.8 | 0.947 | 0.906–0.989* | 0.943 | 0.900–0.988* | 0.854 | 0.808–0.903* | 0.88 | 0.830–0.933* |
| | 240–479 | 18,734 | 22.0 | Ref | | Ref | | Ref | | Ref | |
| | ≥480 | 2,933 | 3.4 | 1.109 | 1.015–1.211* | 1.103 | 1.003–1.212* | 1.018 | 0.910–1.140 | 1.029 | 0.915–1.157 |
| Occasional use | <240 | 9,650 | 11.3 | 0.929 | 0.877–0.984* | 0.97 | 0.911–1.032 | 0.989 | 0.921–1.063 | 1.015 | 0.942–1.094 |
| | 240–479 | 7,634 | 9.0 | 0.953 | 0.896–1.014 | 1.006 | 0.941–1.075 | 1.024 | 0.949–1.106 | 1.013 | 0.935–1.097 |
| | ≥480 | 1,084 | 1.3 | 1.185 | 1.034–1.358* | 1.134 | 0.978–1.316 | 1.043 | 0.875–1.244 | 0.971 | 0.805–1.170 |
| Daily use | <240 | 9,383 | 11.0 | 0.81 | 0.763–0.859* | 0.906 | 0.849–0.966* | 0.887 | 0.824–0.956* | 0.914 | 0.845–0.988* |
| | 240–479 | 8,172 | 9.6 | 0.886 | 0.833–0.942* | 0.986 | 0.923–1.053 | 0.977 | 0.905–1.054 | 0.995 | 0.920–1.077 |
| | ≥480 | 1,297 | 1.5 | 1.031 | 0.906–1.173 | 1.107 | 0.964–1.272 | 0.996 | 0.845–1.173 | 0.999 | 0.843–1.185 |

CI, confidence interval; OR, odds ratio.

[a]Crude is non-adjusted.

[b]Adjusted is adjusted for maternal age, sex, gestational age, education (mother and father), allergy (mother), smoking (during pregnancy), alcohol consumption (during pregnancy), body mass index (before pregnancy), parity, breastfeeding term, nursery school and day-care center for children.

conversely, we found no association between maternal intake of supplements and symptoms in offspring, but high maternal dietary folate intake was associated with wheeze in offspring. However, the definition of exposure in previous reports differed from that in this study. Moreover, to the best of our knowledge, only a few studies have analyzed the risk of symptoms in offspring by maternal dietary folate.

Here, high dietary folate intake was associated with allergies in children aged 2 years, whereas supplement intake was not. Currently, there is no report on the mechanism of allergy development depending on dietary folate or supplement intake. Furthermore, there is no scientific evidence of a biological mechanism of allergies in offspring caused only by dietary folate intake. Therefore, we considered that the intake of other nutrients with folic acid may have contributed to an increased risk of allergies. The BMI of the "≥480 μg/day" group was higher than that of the "<240 μg/day" and "240–479 μg/day" groups (Table 1), indicating a difference in dietary habits and high-calorie food intake. Omega-3 fatty acid and vitamin D intake during pregnancy is perceived to decrease allergies [26,27]. Thus, we examined their contribution and performed further analyses adjusted with omega-3 fatty acids and vitamin D intake determined using the FFQ. Consequently, the OR shifted towards 1 (S2 Table). It remains unclear whether maternal nutrient intake, except for folic acid, increases the risk of symptoms in offspring. A comprehensive analysis is needed to elucidate the association between maternal intake of nutrients and symptoms in offspring.

The major strength of this study is that an extensive study cohort was utilized (~100,000 mothers). Furthermore, 85,114 children were divided into nine groups according to dietary and supplemental folic acid intake by mothers. The amount of consumed dietary folate was based on the FFQ, which is widely used in nutrient intake calculations. Folic acid fortification is not commonly implemented in Japan. Therefore, it is not necessary to consider whether the folic acid values calculated using the FFQ will be misclassified owing to the addition of folic acid supplements. We believe that this was advantageous in this assessment.

However, this study had several limitations. In this study, we excluded 18,948 participants because of the lack of information about folate intake and/or prevalence of wheeze and eczema. Thus, we were not able to compare if there was any bias on folate intake or the prevalence percentage between participants included and excluded. However, we compared basic characteristics such as maternal age, education, as well as the prevalence of wheeze and eczema reported in previous studies of the JECS, indicating that there was no difference in participant characteristics, the prevalence of wheeze and eczema in children, and the percentage of folic acid supplements intake [10,28,29]. Therefore, the analyzed population is not likely to be biased with respect to the original population. Second, maternal folic acid supplement intake information only considers its usage and not its quantity. Third, children in the "wheeze" group exhibited "multipara" and "nursery school and day-care center for children (yes)" (Table 2), probably owing to the confusion between respiratory disease caused by infection from a sibling/friend and wheeze. Forth, genetic factors, such as methylenetetrahydrofolate reductase polymorphism C677T (MTHFR-C677T), were not considered. Maternal MTHFR-C677T may be associated with allergies in offspring through folic acid intake [30]. Finally, there are other risk factors that may cause wheeze and eczema. Potential confounders could not be sufficiently considered in this study, such as indoor and/or outdoor air pollution. For example, previous studies reported that environmental factors such as $PM_{2.5}$ are known risk factors that increase allergies [31–33]. Regarding indoor environments, allergens such as dust mites are well known factors associated with allergies [34], so that these potential confounding factors could be considered in further studies.

## Conclusions

Maternal intake of folic acid supplements is not associated with wheeze and eczema in children at the age of 2. However, high concentrations of dietary folate ($\geq$480 μg/day) increase the risk of wheeze compared with low concentrations (240–479 μg/day) after adjustment. Nonetheless, the mechanisms that explain these results remain unclear; these observations may be attributed to other nutrients and/or calories. This study showed an association between maternal dietary folate intake and allergic symptoms at the age of 2. However, this association represents weak evidence to reconsider intake recommendations. Thus, we suggest the intake of both dietary and supplemental folic acid to avoid neural tube defects. Further studies on the association between the maternal intake of folic acid and allergic symptoms at the age of 3 or above considering potential confounders such as other nutrition, outdoor/indoor air pollution, and genetic factors are needed to confirm the results of this study.

## Supporting information

**S1 Table. Characteristics of the participants (mothers and children).**
(DOCX)

**S2 Table. Folic acid (supplement and diet) and the allergies adjusted with omega-3 fatty acid and vitamin D intake.** †Adjusted is adjusted for maternal age, sex of child, fetus week number, education (mother and father), allergy (mother), smoking (during pregnancy), alcohol consumption (during pregnancy), body mass index (before pregnancy), parity, breastfeeding term, nursery school and baby farm, omega-3 fatty acid intake ($<$ 1.6 g/day, $\geq$ 1.6 g/day), vitamin D intake ($<$ 8.5 μg/day, $\geq$ 8.5 μg/day). The intake of omega-3 fatty acid and vitamin D was classified based on the maternal standard issued by the Ministry of Health, Labor, and Welfare in Japan (2020). ‡OR, odds ratio; §CI, confidence interval.
(DOCX)

## Acknowledgments

We would like to thank all study participants and the director of the Programme Office. We would also like to thank Editage (www.editage.com) for English language editing.

Members of the JECS Group as of 2021: Michihiro Kamijima (principal investigator, Nagoya City University, Nagoya, Japan E-mail: kamijima@med.nagoya-cu.ac.jp), Shin Yamazaki (National Institute for Environmental Studies, Tsukuba, Japan), Yukihiro Ohya (National Center for Child Health and Development, Tokyo, Japan), Reiko Kishi (Hokkaido University, Sapporo, Japan), Nobuo Yaegashi (Tohoku University, Sendai, Japan), Koichi Hashimoto (Fukushima Medical University, Fukushima, Japan), Chisato Mori (Chiba University, Chiba, Japan), Shuichi Ito (Yokohama City University, Yokohama, Japan), Zentaro Yamagata (University of Yamanashi, Chuo, Japan), Hidekuni Inadera (University of Toyama, Toyama, Japan), Takeo Nakayama (Kyoto University, Kyoto, Japan), Hiroyasu Iso (Osaka University, Suita, Japan), Masayuki Shima (Hyogo College of Medicine, Nishinomiya, Japan), Youichi Kurozawa (Tottori University, Yonago, Japan), Narufumi Suganuma (Kochi University, Nankoku, Japan), Koichi Kusuhara (University of Occupational and Environmental Health, Kitakyushu, Japan), and Takahiko Katoh (Kumamoto University, Kumamoto, Japan).

## Author Contributions

**Conceptualization:** Hideyuki Masuda, Sumitaka Kobayashi, Chihiro Miyashita, Sachiko Itoh, Yu Ait Bamai, Yasuaki Saijo, Yoshiya Ito, Reiko Kishi, Atsuko Ikeda-Araki.

**Data curation:** Hideyuki Masuda, Sumitaka Kobayashi, Chihiro Miyashita, Sachiko Itoh, Yu Ait Bamai, Yasuaki Saijo, Yoshiya Ito, Atsuko Ikeda-Araki.

**Formal analysis:** Hideyuki Masuda, Atsuko Ikeda-Araki.

**Investigation:** Chihiro Miyashita, Sachiko Itoh, Yu Ait Bamai, Yoshiya Ito, Reiko Kishi, Atsuko Ikeda-Araki.

**Methodology:** Sumitaka Kobayashi, Atsuko Ikeda-Araki.

**Visualization:** Hideyuki Masuda, Sachiko Itoh, Yasuaki Saijo, Atsuko Ikeda-Araki.

**Writing – original draft:** Hideyuki Masuda.

**Writing – review & editing:** Sumitaka Kobayashi, Chihiro Miyashita, Sachiko Itoh, Yu Ait Bamai, Yasuaki Saijo, Yoshiya Ito, Reiko Kishi, Atsuko Ikeda-Araki.

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
