## [Decision Letter · Decision Letter 0]

10 May 2022

PONE-D-22-10468Maternal dietary folate intake with folic acid supplement use and childhood wheeze and eczema in the Japan Environment and Children's StudyPLOS ONE

Dear Dr. Araki,

Thank you for submitting your manuscript to PLOS ONE. After careful consideration, we feel that it has merit but does not fully meet PLOS ONE’s publication criteria as it currently stands. Therefore, we invite you to submit a revised version of the manuscript that addresses the points raised during the review process.

Numerous advantages of the authors’ manuscript over the studies so far published include the largest-ever sample size and the full consideration of potential confounders, which was highly appreciated by Reviewer 3. Taken together, I would like to encourage the authors to revise the manuscript while making clearer the following points that the three reviewers have kindly raised. Let me summarize as follows.

1. Perhaps the significance of the findings is more strongly upheld if the authors discuss more on the rationale of, and the consequences of, the adjustment for numerous covariates (Reviewer 1). Confounding by maternal education and smoking may be of a particular relevance with this regard considering the points addressed in the paper recommended by Reviewer 2.

2. Please rephrase the hypothesis the authors posed; “maternal intake of supplement along with dietary folate intake” is a bit too vague. I suppose that, because of this unclarity, Reviewer 1 has felt that readers are not carefully guided to the analysis plan. Also, the additional analysis (LL. 234 and onward) sounds abrupt to me as this analysis was not originally planned.

3. Please re-check the consistency and clarity of the text (Reviewer 2). An example include “however” in LL 225 and “however” in the following sentence.

We look forward to receiving your revised manuscript.

Kind regards,

Kenji J Tsuchiya, MD, PhD

Academic Editor

PLOS ONE

Journal Requirements:

3.  One of the noted authors is a group or consortium "Japan Environment and Children’s Study (JECS) Group". In addition to naming the author group, please list the individual authors and affiliations within this group in the acknowledgments section of your manuscript. Please also indicate clearly a lead author for this group along with a contact email address.

Reviewers' comments:

Reviewer's Responses to Questions

**Comments to the Author**

1. Is the manuscript technically sound, and do the data support the conclusions?

Reviewer #1: No

Reviewer #2: Yes

Reviewer #3: Yes

2. Has the statistical analysis been performed appropriately and rigorously? 

Reviewer #1: Yes

Reviewer #2: Yes

Reviewer #3: Yes

3. Have the authors made all data underlying the findings in their manuscript fully available?

Reviewer #1: Yes

Reviewer #2: No

Reviewer #3: Yes

4. Is the manuscript presented in an intelligible fashion and written in standard English?

Reviewer #1: Yes

Reviewer #2: Yes

Reviewer #3: Yes

5. Review Comments to the Author

Reviewer #1: The long effect of maternal intake of dietary folate and folic acid supplement on health of offspring is worthy of assessment because folic acid supplement has become popular for pregnant women around the world. Unfortunately, the evidence from the population remains limited. This study focused on maternal dietary folate intake with folic acid supplement use and childhood wheeze and eczema, and found something interesting. However, some technical issues should be addressed further.

1. It is unclear why the authors selected wheeze and eczema as indicator for allergy. Maybe more review should be included in the part of introduction.

2. An important technical flaw is that authors did not provide strategy of statistical analysis for data in details. If possible, add it please.

3. Due to lots of potential confounders, how did authors consider them? It could be one of limitation of this study.

4. The data used here was from a cohort project. Lots of participants were excluded due to various reasons. Is there any difference between the participants included and those excluded? If yes, whether is the association biased? Maybe something more should be done about this issue.

5. In this study, whether or not other allergens during childhood are considered, which could confound this association between maternal folate or folic acid and allergy of offspring. Pls say something more in discussion.

6. Conclusion should be done with caution because of confounders and weak association from the table 4.

Reviewer #2: In general, this is a good manuscript aiming to understand the relationship between folic acid intake, wheeze and eczema. The authors used data from nearly 100,000 mother-child pairs whose folic acid intake was self-reported. This is a main limitiation of the study that should be stressed.

Please give more details on the instrument for dietary data collection (FFQ)

Please consider, if possible, the adherence to recommendation on folic acid use during pregnancy. Have the authors information on the timing of folic acid intake (before pregnancy or after the conception). This is also important to understand the effect of the duration of folic acid intake. (please consider the following 10.3390/ijerph17020638).

I would also suggest a double-check of the text for revising minor errors and typos.

Reviewer #3: This is an excellent work. The authors found that high maternal intake of dietary folate was a risk factor for wheeze and eczema in children at 2 years of age, but folic acid supplements were not associated with wheeze and eczema in the offspring. The strength of this work is the nationwide birth cohort study with huge sample with 84,361 mothers and 85,114 children. Some minor suggestions or comments are as follows:

1. TITLE. I suggest the authors to modify the current title to a more accurate and attractive one such as "Maternal dietary folate intake with folic acid supplement use and wheeze and eczema in children aged 2 years in Japan: A nationwide cohort study".

2. About the limitations in DISCUSSION. Another two aspects should be mentioned:

(1) The authors didn't consider the indoor and outdoor environmental factors. The authors mentioned "These factors (covariates) were selected based on previous studies that reported an association between folic acid intake during pregnancy and allergies in offspring", but the environmental factors have been widely considered to be the risk factors for childhood allergic sympotms/diseases. Some references include:

-- Onset and remission of childhood wheeze and rhinitis across China - associations with early life indoor and outdoor air pollution. Environment International 2019, 123: 61-69.

-- Preconceptional, prenatal and postnatal exposure to outdoor and indoor environmental factors on allergic diseases/symptoms in preschool children. Chemosphere 2016, 152: 459-467

(2) As the authors mentioned in ABSTRACT that "The study does not provide evidence that pregnant women should be denied folic acid intake owing to the increased risk early onset allergies in offspring", further/future studies focusing on the its association with allergic diseases, asthma or rhinitis, in elder preschool children aged 3-6 years are necessary to confirm the conclusions of the present work.

6. PLOS authors have the option to publish the peer review history of their article (what does this mean?). If published, this will include your full peer review and any attached files.

Reviewer #1: No

Reviewer #2: No

Reviewer #3: No

---

## [Author Response · Author response to Decision Letter 0]

21 Jun 2022

We would like to thank you for the reviewers for the valuable comments that have helped us to substantially improve the manuscript. For a point-by-point response to these comments, please refer to the followings, and the text marked in yellow is the corrected part. Pages and lines are referring clean version of the manuscript.

RESPONSES TO THE COMMENTS OF REVIEWER 1

1. It is unclear why the authors selected wheeze and eczema as indicator for allergy. Maybe more review should be included in the part of introduction.

Response: We thank you for this pertinent comment.

In previous studies, allergic symptoms such as asthma, wheeze, and atopic dermatitis have been associated with the maternal intake of folic acid. Therefore, we used wheeze and eczema, which are early symptoms of asthma and atopic dermatitis, as the outcomes. 

Per your comment, we have revised the following text in the Introduction (p. 5, line 103): 

In previous studies, allergic symptoms such as asthma, wheeze, and atopic dermatitis in offspring were associated with the maternal intake of folic acid [14–18]. In this study, we evaluated the association between folic acid supplements and dietary folate intake during pregnancy and wheeze and eczema, which are early symptoms of asthma and atopic dermatitis in offspring.

2. An important technical flaw is that authors did not provide strategy of statistical analysis for data in details. If possible, add it please.

Response: In accordance with your comment, we have revised the following text in the Material and Methods (p. 7, line 151): First, we examined the associations with folic acid supplement and dietary folate intake separately. For folic acid supplement intake, “no use” was selected as a reference, whereas for dietary folate intake, “240–479 µg/day” was selected as the reference range based on the recommended consumption of folate per day for individuals during the first trimester (240 µg/day) and second/third trimester (480 µg/day) of pregnancy [6]. Second, to examine the interaction of maternal folic acid supplementation and dietary folate intake, we combined folic supplementation and dietary folate intake to find the association with symptoms among offspring. In this analysis, “no use” and” 240–479 µg/day” were selected as the reference [8].

3. Due to lots of potential confounders, how did authors consider them? It could be one of limitation of this study.

Response: There may have been potential confounders such as maternal/paternal genetic factors, indoor/outdoor air pollution, and other allergens. We have added the following limitation in the Discussion.

(p. 15, line 297): Finally, there are other risk factors that may cause wheeze and eczema. Potential confounders could not be sufficiently considered in this study, such as indoor and/or outdoor air pollution. For example, previous studies reported that environmental factors such as PM2.5 are known risk factors that increase allergies [31-33]. Regarding indoor environments, allergens such as dust mites are well known factors associated with allergies [34], so that these potential confounding factors could be considered in further studies.

4. The data used here was from a cohort project. Lots of participants were excluded due to various reasons. Is there any difference between the participants included and those excluded? If yes, whether is the association biased? Maybe something more should be done about this issue.

Response: We excluded 18,948 participants because of the lack of information about folate intake and/or prevalence of wheeze and eczema. Thus, we were not able to compare if there was any bias of folate intake or a prevalence percentage between participants included and excluded. However, there was no difference in participant characteristics, the prevalence of wheeze, and eczema in children, and the percentage of folic acid supplement intake was compared with that of previous studies of JECS. We have added the following sentence to the limitations.

(p. 15, line 282): In this study, we excluded 18,948 participants because of the lack of information about folate intake and/or prevalence of wheeze and eczema. Thus, we were not able to compare if there was any bias on folate intake or the prevalence percentage between participants included and excluded. However, we compared basic characteristics such as maternal age, education, as well as the prevalence of wheeze and eczema reported in previous studies of the JECS, indicating that there was no difference in participant characteristics, the prevalence of wheeze and eczema in children, and the percentage of folic acid supplements intake [10, 28, 29]. Therefore, the analyzed population is not likely to be biased with respect to the original population. 

5. In this study, whether or not other allergens during childhood are considered, which could confound this association between maternal folate or folic acid and allergy of offspring. Pls say something more in discussion.

Response: As per our reply to your 3rd comment, we have added the following sentences to the limitations regarding other allergens during childhood.

(p. 15, line 297): Finally, there are other risk factors that may cause wheeze and eczema. Potential confounders could not be sufficiently considered in this study, such as indoor and/or outdoor air pollution. For example, previous studies reported that environmental factors such as PM2.5 are known risk factors that increase allergies [31-33]. Regarding indoor environments, allergens such as dust mites are well known factors associated with allergies [34], so that these potential confounding factors could be considered in further studies.

6. Conclusion should be done with caution because of confounders and weak association from the table 4.

Response: There was the possibility that some confounders affected the results shown in Table 4. In further studies, more confounders such as nutrients, environmental, and genetic factors must be considered. We have added the following sentences to the conclusion. 

(p. 16, line 312): However, this association represents weak evidence to reconsider intake recommendations. Thus, we suggest the intake of both dietary and supplemental folic acid to avoid neural tube defects. Further studies on the association between the maternal intake of folic acid and allergic symptoms at the age of 3 or above considering potential confounders such as other nutrition, outdoor/indoor air pollution, and genetic factors are needed to confirm the results of this study.

RESPONSES TO THE COMMENTS OF REVIEWER ２

We thank you for the insightful comments, which have helped us to substantially improve the manuscript.

1. Please give more details on the instrument for dietary data collection (FFQ) 

Response: We thank you for this pertinent comment.

In accordance with the comment, we have added the following text to the Materials and Methods (p. 6, line 134): In brief, the FFQ asked the participants about the frequency of consumption of 172 food and beverage items, from never to seven or more times per day for food, and ten or more glasses per day for beverages. Thereafter, the intake of 53 nutrients was calculated. 

2. Please consider, if possible, the adherence to recommendation on folic acid use during pregnancy. Have the authors information on the timing of folic acid intake (before pregnancy or after the conception). This is also important to understand the effect of the duration of folic acid intake. (please consider the following 10.3390/ijerph17020638). I would also suggest a double-check of the text for revising minor errors and typos.

Response: Thank you for your comment. We agree with your recommendation and have added the following sentences to the conclusion. (p. 16, line 312): “However, this association represents weak evidence to reconsider intake recommendations. Thus, we suggest the intake of both dietary and supplemental folic acid to avoid neural tube defects.” We could not consider the timing and duration of folic acid intake, which is an important issue to consider in future research.

In addition, we thank you for your suggestion to double-check the manuscript. We rechecked the manuscript and fixed grammatical and typographical errors as needed. 

RESPONSES TO THE COMMENTS OF REVIEWER 3

We thank you for the insightful comments, which have helped us to substantially improve the manuscript. 

1. TITLE. I suggest the authors to modify the current title to a more accurate and attractive one such as "Maternal dietary folate intake with folic acid supplement use and wheeze and eczema in children aged 2 years in Japan: A nationwide cohort study".

Response: We thank you for this pertinent comment.

In accordance with the comment, we have revised the title as follows: 

"Maternal dietary folate intake with folic acid supplements and wheeze and eczema in children aged 2 years in the Japan Environment and Children’s Study." As a part of a nationwide cohort study, we were recommended to include “Japan Environment and Children’s Study” in the title.

2. About the limitations in DISCUSSION. Another two aspects should be mentioned: (1) The authors didn't consider the indoor and outdoor environmental factors. The authors mentioned "These factors (covariates) were selected based on previous studies that reported an association between folic acid intake during pregnancy and allergies in offspring", but the environmental factors have been widely considered to be the risk factors for childhood allergic sympotms/diseases. Some references include: -- Onset and remission of childhood wheeze and rhinitis across China - associations with early life indoor and outdoor air pollution. Environment International 2019, 123: 61-69. -- Preconceptional, prenatal and postnatal exposure to outdoor and indoor environmental factors on allergic diseases/symptoms in preschool children. Chemosphere 2016, 152: 459-467

Response: There may have been potential confounders such as maternal/paternal genetic factors, indoor/outdoor air pollution, and other allergens. We have added the following limitation to the Discussion.

(p. 15, line 297): Finally, there are other risk factors that may cause wheeze and eczema. Potential confounders could not be sufficiently considered in this study, such as indoor and/or outdoor air pollution. For example, previous studies reported that environmental factors such as PM2.5 are known risk factors that increase allergies [31-33]. Regarding indoor environments, allergens such as dust mites are well known factors associated with allergies [34], so that these potential confounding factors could be considered in further studies.

(2) As the authors mentioned in ABSTRACT that "The study does not provide evidence that pregnant women should be denied folic acid intake owing to the increased risk early onset allergies in offspring", further/future studies focusing on the its association with allergic diseases, asthma or rhinitis, in elder preschool children aged 3-6 years are necessary to confirm the conclusions of the present work.

Response: We agree with the comment. We have incorporated this suggestion throughout the abstract and conclusion. 

(p. 2, line 42): In contrast, only dietary folate intake was positively associated with wheeze at the age of 2 (adjusted odds ratio, 1.103; 95% confidence interval, 1.003–1.212). However, there is no scientific evidence of a biological mechanism that clarifies this result. Potential confounders such as other nutrition, outdoor/indoor air pollution, and genetic factors may have affected the results. Therefore, further studies on the association between maternal intake of folic acid and allergic symptoms at the age of 3 or above are needed to confirm the results of this study.

(p. 16, line 313): Thus, we suggest the intake of both dietary and supplemental folic acid to avoid neural tube defects. Further studies on the association between the maternal intake of folic acid and allergic symptoms at the age of 3 or above considering potential confounders such as other nutrition, outdoor/indoor air pollution, and genetic factors are needed to confirm the results of this study.

---

## [Decision Letter · Decision Letter 1]

29 Jul 2022

Maternal dietary folate intake with folic acid supplements and wheeze and eczema in children aged 2 years in the Japan Environment and Children’s Study

PONE-D-22-10468R1

Dear Dr. Araki,

We’re pleased to inform you that your manuscript has been judged scientifically suitable for publication and will be formally accepted for publication once it meets all outstanding technical requirements.

Kind regards,

Kenji J Tsuchiya, MD, PhD

Academic Editor

PLOS ONE

Additional Editor Comments (optional):

Thank you very much for addressing all the questions and comments the reviewers have raised. They are satisfied with your replies.

Reviewers' comments:

Reviewer's Responses to Questions

**Comments to the Author**

1. If the authors have adequately addressed your comments raised in a previous round of review and you feel that this manuscript is now acceptable for publication, you may indicate that here to bypass the “Comments to the Author” section, enter your conflict of interest statement in the “Confidential to Editor” section, and submit your "Accept" recommendation.

Reviewer #1: All comments have been addressed

Reviewer #2: All comments have been addressed

Reviewer #3: All comments have been addressed

2. Is the manuscript technically sound, and do the data support the conclusions?

Reviewer #1: Yes

Reviewer #2: Yes

Reviewer #3: Yes

3. Has the statistical analysis been performed appropriately and rigorously? 

Reviewer #1: Yes

Reviewer #2: Yes

Reviewer #3: Yes

4. Have the authors made all data underlying the findings in their manuscript fully available?

Reviewer #1: Yes

Reviewer #2: Yes

Reviewer #3: Yes

5. Is the manuscript presented in an intelligible fashion and written in standard English?

Reviewer #1: Yes

Reviewer #2: Yes

Reviewer #3: Yes

6. Review Comments to the Author

Reviewer #1: (No Response)

Reviewer #2: (No Response)

Reviewer #3: This manuscript was improved according to the reviewers' comments. The quality is now fine. I suggest to accept this work for publication.

7. PLOS authors have the option to publish the peer review history of their article (what does this mean?). If published, this will include your full peer review and any attached files.

Reviewer #1: No

Reviewer #2: No

Reviewer #3: No

---

## [Editor Report · Acceptance letter]

12 Aug 2022

PONE-D-22-10468R1 

Maternal dietary folate intake with folic acid supplements and wheeze and eczema in children aged 2 years in the Japan Environment and Children’s Study 

Dear Dr. Ikeda-Araki:

I'm pleased to inform you that your manuscript has been deemed suitable for publication in PLOS ONE. Congratulations! Your manuscript is now with our production department. 

Kind regards, 

on behalf of

Dr. Kenji J Tsuchiya 

Academic Editor

PLOS ONE